# Nutritional Compositions of Aquatic Insects Living in Rice Fields, with a Particular Focus on Odonate Larvae

**DOI:** 10.3390/insects13121131

**Published:** 2022-12-07

**Authors:** Witwisitpong Maneechan, Akekawat Vitheepradit, Taeng On Prommi

**Affiliations:** 1Program of Bioproducts Science, Department of Science, Faculty of Liberal Arts and Science, Kasetsart University, Kamphaeng Saen Campus, Nakhon Pathom 73140, Thailand; 2Department of Entomology, Kasetsart University, Bangkok 10900, Thailand; 3Department of Science, Faculty of Liberal Arts and Science, Kasetsart University, Kamphaeng Saen Campus, Nakhon Pathom 73140, Thailand

**Keywords:** aquatic insect, nutritional composition, Odonata, heavy metal, microplastic

## Abstract

**Simple Summary:**

Food security concerns are growing due to the rapid increase in the world population. From this perspective, insects are a possible sustainable food source because of their nutritional value and the sustainability of their production system. Although the human consumption of edible insects has been a culturally long-standing practice, the nutritional literature on aquatic insects is not complete. Thus, the aims of the present study were to: (1) confirm the nutritional characteristics of odonate larvae (Libellulidae: *Pantala* sp.), including quantifying the bioaccumulation; and (2) investigate the microplastic accumulation in odonate larvae living in rice fields. The results show that odonates such as *Pantala* sp. are a good source of protein, minerals, essential amino acids, and long-chain polyunsaturated fatty acids. However, although the odonates seem to be a good source of nutrition, they may typically contain bioaccumulation, including microplastics, from their diets and habitats.

**Abstract:**

Although the human consumption of aquatic insects is prevalent in many regions, the nutritional composition of the insects has not been comprehensively determined. The proximate composition of *Pantala* sp. was shown to be a good source of protein (49.45 ± 0.32 g/100 g DW), as well as of minerals such as sodium, calcium, potassium, phosphorus, zinc, and iron. All nine essential amino acids are present in this species, with valine being the most abundant. The major fatty acids are palmitic acid (1.19 ± 0.02 g/100 g DW), oleic acid (0.63 ± 0.02 g/100 g DW), and linoleic acid (0.55 ± 0.01 g/100 g DW). Lead (Pb), arsenic (As), and cadmium (Cd) showed a value of 0.18 ± 0.01 mg·kg^−1^, 3.51 ± 0.12 mg·kg^−1^, and 0.17 ± 0.00 mg·kg^−1^, respectively. Furthermore, microplastic (MP) contamination in odonate larvae (419 individuals belonging to three identified families) was found in varying shapes, e.g., fibers, fragments, and rods. FTIR analysis revealed the following MP polymers, polyethylene terephthalate, polyvinyl acetate, bis(2-ethylhexyl), polybutadiene, poly(methyl methacrylate-co-methacrylic acid); P(MMA-co-MA), poly(ethylene glycol) tetrahydrofurfuryl ether, poly(acrylonitrile-co-butadiene), and polypropylene glycol. The results of this work could be a nutritional reference for food security and the risk of eating insects.

## 1. Introduction

Given that the world population is expected to rapidly increase to 9.6 billion people by 2050, there are increasing concerns about food security [1]. From this perspective, edible insects are considered to contribute to the food security of the world [2], due to their nutritional value and the sustainability of their production system. Globally, about 14 insect orders or 2000 insect species contain edible insects [3]. Because they can live in and adapt to a variety of air, water, and land habitats, they are the most numerous and diverse groups in the ecosystem. They also have a strong reproductive capacity [4]. Consuming insects, also known as entomophagy, is a good source of amino acids and fatty acid profiles as well as protein (20–76% of dry matter) and fat (2–50% of dry matter) [5]. Beetles (Coleoptera) (31%) are the most widely consumed edible insects in the world, followed by caterpillars (Lepidoptera) (18%), bees, wasps, and ants (Hymenoptera) (14%) [6]. Many studies have recorded that aquatic insects are used as food, but their nutritional value and bioaccumulation, including microplastics (MPs), have received less attention.

Aquatic insects have a very wide species diversity. Only 12 orders and 10% of insect species are classified as aquatic insects [7,8]. Taxonomically, they belong to several of the same orders as terrestrial insects. Williams and Williams [7] have summarized the relevant biology, natural habitats, and comparisons of aquatic insect orders. Candidate species for food and feed are most likely to be found in six of the twelve orders of aquatic insects [7,9,10]. They include Coleoptera (beetles), Diptera (true flies), Ephemeroptera (mayflies), Hemiptera (true bugs), Odonata (dragonflies and damselflies), and Trichoptera (caddisflies). The order Megaloptera has the potential to be a candidate species, according to study and utilization reports from Southwest China and Japan, because it has long been used as food and folk medicine and has significant economic value [11,12]. Insects are distributed throughout the world, and certain species’ breeding techniques have become more efficient [13]. The methods provided by Macadam and Stockan [10] have been used to update the list of edible aquatic insects. The list contains 329 species from 153 genera and 51 families, coming from 46 countries. Edible aquatic insects make up approximately 15% of all insect species, of which, over 2000 are consumed. The species are classified into eight orders, from highest to lowest: Coleoptera, Odonata, Hemiptera, Diptera, Trichoptera, Megaloptera, Ephemeroptera, and Plecoptera. Over 3/4 of the species belong to the three orders: Coleoptera, Odonata, and Hemiptera, and they are all predators. The main edible stage of aquatic insects such as dragonflies (Order Odonata) is the naiad, or larva, because it is easier to capture than the adult form. The previous study shows that dragonfly nymphs are commonly consumed in China, India, the Philippines, Laos, and Thailand [14,15,16,17,18]. Several researchers have previously documented the nutritional makeup of Odonata insects [5,14,19]. Due to the fact that most aquatic insect larvae or naiads are physically indistinguishable most of the time, the number of edible aquatic insect species may be underestimated [10]. With the aid of molecular biology techniques, an increasing number of species of edible aquatic insects will be identified.

The ecosystems of rice fields serve as temporary clearly manmade wetlands that are associated with or share water with natural wetlands [20]. Today, agricultural activities, including pesticides, fertilizers, and multiple uses of water, are thus considered important pools of heavy metal contaminants in agricultural products and aquatic ecosystems. Pesticides and fertilizers contain cadmium (Cd), lead (Pb), and arsenic (As) [21]. These heavy metals can give rise to the bioaccumulation of toxicity in aquatic species. For instance, aquatic insects can accumulate elements directly from the sediment, water, or food, based on their lifecycle [22]. In this study, we focus on those species which are commonly consumed by humans, such as odonate larvae. This is of concern because they are associated with harm to human health, due to the many aquatic insects being reported as food [9,10,23]. A risk factor for the accumulation of heavy metals is the long-term negative impact on human health, affecting the brain or causing cancer or death from exposure to high levels [24,25]. Hence, the World Health Organization [26] considers As, Cd, and Pb as hazardous chemicals. In the same way, microplastics (MPs) are of concern because they can cause significant environmental impacts and a risk to human health if incorporated into the food chain [27]. Microplastics are small plastic particles that range in size from 1 μm to 5 mm [28]. These may degrade from primary MPs (manufactured particles) or secondary MPs (derived by physical, chemical, and/or biological degradation from larger plastics) and can be efficiently dispersed in the natural environment [29]. A variety of aquatic animals can either ingest MPs from the bottom sediment, suspended particulates, or their diets, which contain MP particle contamination [30]. The monitoring of these contaminants can provide useful information for evaluating the contamination in aquatic ecosystems. Thus, the aims of the study were: (1) to confirm the nutritional characteristics of odonate larvae, *Pantala* sp., including quantifying the bioaccumulation; and (2) to investigate the MP accumulation in odonate larvae living in rice fields.

## 2. Materials and Methods

### 2.1. A Sampling of Aquatic Insects

This study was carried out at five sampling rice fields, located in Nakhon Pathom Province, in the central part of Thailand (Figure 1). Aquatic insects were collected using an aquatic net, approximately 100 m along the edge of the rice plots. The collected organisms were placed in white trays for sorting, and the captured odonate specimens (Figure 2) were transferred to containers for identification in the laboratory. Three replicate samples were collected at each site. Using taxonomic keys, aquatic insects were identified under a stereo microscope (Olympus SZ51) [31,32]. Then, the odonate samples were pooled (a total of 419 individuals) by taxon for microplastic content determination. Moreover, the odonate specimens were collected for nutritional analysis, with a focus on *Pantala* sp.

### 2.2. Sample Preparation for Nutrition Analysis

Specimens of aquatic dragonfly nymphs (*Pantala* sp.), 3447 mostly final instar nymphs, were rinsed with clean running water and sun-dried. The final instars have been described as having long spread apart wing buds. For analysis, approximately 50 g (three replicates) of the dried specimens were used. The dried specimens were kept cool in a freezer for further biochemical analysis. Each parameter of the biochemical analysis was determined three times, and the results are reported as the mean ± standard deviation. The proximate analysis was estimated following the Association of Official Analytical Chemists [33] to determine the moisture (AOAC method 925.45), protein (AOAC method 991.20), and ash content (AOAC method 923.03). The fat content (AOAC method 2003.05) was determined by the Soxhlet methods, according to AOAC (2019) [34]. Inductively coupled plasma optical emission spectroscopy (ICP-OES) was used to determine the mineral and toxic heavy metal concentrations. High-performance liquid chromatography (HPLC-Agilent 1260 Infinity series, fluorescence detector) was used to determine the amino acid composition. An Agilent 7890B gas chromatograph (GC) and modified Compendium of Methodology for Food Analysis methods (2003) were used to determine the fatty acid profile.

### 2.3. Microplastics Extraction and Identification

All the specimens of odonate larvae were rinsed three times with distilled water to remove a variety of contaminants, after which each species was pooled and transferred into glass beakers. For the digestion of organic matter, 20 mL of hydrogen peroxide solution (30% H_2_O_2_) was added to each beaker, which was covered immediately in parafilm. After that, the beakers were placed in a shaking water bath at 150 rpm and 60 °C for 4 h. Following tissue disintegration, the MP particles were isolated from the remaining matrix by density floatation using 1.6 g/mL potassium formate solution and filtered onto membrane filters (pore size of 0.45 µm; diameter of 47 mm) using a vacuum pump. The filter papers were then placed in clean Petri dishes, capped with aluminum foil, and dried at 50 °C in a drying oven. Each paper was then visually examined and imaged using a Leica EZ4E Stereomicroscope to identify the MP particles based on their size, color, and shape. Selected particles were analyzed to determine the types of polymers commonly present using Fourier transform infrared spectrometer (FTIR) in attenuated total reflection (ATR) mode. The spectra of the polymers were compared to the Bruker spectrum library, with a quality index ≥0.7 being accepted [35].

Regarding contamination control, exclusive gloves (nitrile) and glass and metal ware were used at the laboratory. All the glass and metal were rinsed three times with deionized water and were immediately covered with aluminum foil when not in use. All the experimental procedures were finished as soon as possible.

### 2.4. Statistical Analysis

Each experiment was performed thrice, and the data were analyzed using OriginPro v8 (© OriginLab Corporation, Northampton, MA, USA). All data were represented as mean ± standard deviation (SD).

## 3. Results and Discussion

### 3.1. Proximate Composition

Table 1 shows the proximate composition of the odonate larvae, *Pantala* sp., living in the rice field. These aquatic insects were high in protein content (49.45 ± 0.32 g/100 g DW). Indeed, aquatic insects such as Odonata tend to be excellent sources of protein, 40–65% [7]. However, the data on the protein content of *Pantala* sp. were lower compared to the protein values of *Sympetrum* sp. (Odonata: Libellulidae) reported by Narzari and Sarmah [5]. Similarly, the level of fat content in this dragonfly species was lower compared to two edible insect species that are consumed in Assam, India [36]. Regarding the carbohydrate content of the studied insects (8.80 g/100 g), it was rather low compared with the main sources of energy for humans and animals. The macronutrient (protein and fat) composition in these edible insects reflects their gross energy value, which is also influenced by other factors such as diet and sex [37]. According to Kouřimská and Adámková [38], larvae are usually richer in energy compared to the adults. Conversely, high-protein insect species have lower energy content.

### 3.2. Fatty Acid Composition

The fatty acid composition of the odonate larvae, *Pantala* sp., is summarized in Table 2. Eighteen fatty acids were detected in this study, including seven saturated fatty acids (SFAs), three monounsaturated fatty acids (MUFAs), and seven polyunsaturated fatty acids (PUFAs). Aquatic insects are a rich source of SFAs, MUFAs, and PUFAs [23]. In this study, palmitic acid and stearic acid were the main saturated fatty acids detected. These findings are similar to those reported previously in black soldier flies, *C. vomitoria*, *A. domesticus*, and *R. nitidula* by Bbosa et al. [39]. The main monounsaturated fatty acid (MUFA) was oleic acid, contributing 0.63 g/100 g. Among the polyunsaturated fatty acids (PUFAs) in this study, linoleic acid (LA, 18:2n-6) and arachidonic acid (AA, 20:4n-6) were the two most abundant n-6 PUFAs, whereas alpha-Linolenic acid (ALA, C18:3n3) and eicosapentaenoic acid (EPA, C20:5n3) were the two most abundant n-3 PUFAs. SFAs are a dietary factor with the greatest negative effect on LDL cholesterol. On the contrary, MUFAs such as oleic acid are known to lower plasma cholesterol concentrations [40]. EPA can reduce the risk of cardiovascular disease in humans [41].

When compared with that reported in previous studies [40], this odonate larva species in rice fields contained an EPA higher than that in the five terrestrial insects consumed in South Korea. Long-chain PUFA, particularly omega-3 fatty acids, are commonly found in aquatic organisms, i.e., aquatic insects, freshwater fish, etc., which may be due to their aquatic lifecycle stage. Aquatic insects obtain long-chain PUFA from their diets, i.e., small organisms and algae living in water, and they may possibly synthesize them using the delta-5 and delta-6 desaturase enzymes [42,43].

### 3.3. Amino Acids Composition

The amino acid composition is shown in Table 3.

The essential amino acids (EAAs) were present in this species. Among the essential amino acids, valine was the most abundant. Of the non-essential amino acids, arginine was the most abundant. This aquatic insect species contained significantly more essential amino acids than conventional animal sources. As an example, *Pantala* sp. contained 1.23 g/100 g of histidine, 2.24 g/100 g of lysine, and 0.85 g/100 g of methionine, which is more valuable than beef, pork, and chicken, as reported by Longvah et al. [44] and Amadi and Kiin-Kabari [45]. Thus, these amino acid results indicate that the odonate larvae, *Pantala* sp., are rich in essential amino acids critical for humans.

### 3.4. Mineral Contents and Heavy Metals of the Odonate Larvae (Pantala sp.)

The minerals and heavy metals are shown in Table 4. The odonate larvae, *Pantala* sp., appear to be a good source of sodium, calcium, potassium, and phosphorus. The iron content of 86.74 mg/100 g in *Pantala* sp. was higher than the 13.50 mg/100 g reported in *Sympetrum* sp. (Odonata: Libellulidae) [5]. Earlier studies in western Kenya found insects to be rich in iron (18–1562 mg/100 g dry matter), including ants, termites, and crickets [46]. The high zinc content was 6.18 mg/100 g, which was higher than the 1.69–5.66 mg/100 g reported by Das and Hazarika [47] in edible Coleopteran. Iron and zinc are of significance, especially considering the importance of these trace elements in health and nutrition. Iron and zinc deficiency represent major public health problems that affect human health [48]. Thus, the presence of individual minerals, i.e., sodium, calcium, potassium, phosphorus, zinc, and iron, was sufficient for human nutrition. However, it is impossible to rule out the possibility that soil contamination, particularly from benthic macroinvertebrates, may have resulted in an overestimation of mineral content. Because of the close relationship between aquatic species and their habitat, aquatic insects easily accumulate various pollutants. The analysis of heavy elements in *Pantala* sp. indicated that, among the three heavy elements tested, only arsenic had a high concentration. The mean heavy element concentration of the insect samples is given in Table 4. Cd and Pb in *Pantala flavescens* were reported from wetlands in Iran [49], which were low, as those in Table 4, although our Cd values were higher. Similarly, Simon et al. [50] reported higher Pb levels in Odonate (*Gomphus flavipes*) larvae (as compared to Table 4) for rivers in Hungary impacted by mine spills. In our study, the arsenic concentrations were relatively high, which, even so, were less than those found by Addo-Bediako and Malakane [51]. Similarly, Aydogan et al. [22] also reported high levels of arsenic in some aquatic insects (Coleoptera) (0.2–14.4 mg·kg^−1^) as well. In aquatic insects, the concentrations of metals (i.e., Cd, Ni, Cr, As, Pb, Cu, Ti, Zn, and Mn) differ with the size, lifecycle stages, and different bioaccumulation patterns [52]. From the results of the heavy metal contamination in *Pantala* sp., the order of the mean concentration of heavy metals was As > Pb > Cd. Thailand has not established maximum levels for heavy metals in insect products. However, the Pb and Cd content in odonate *Pantala* sp. did not exceed the permissible limit of the Codex Alimentarius International Food Standards [53] when compared with marine products (fish, bivalves, mollusks, and cephalopods). On the other hand, the As content of the analyzed samples showed that the levels of 3.51 ± 0.12 mg·kg^−1^ exceeded the permissible limits (2.0 mg·kg^−1^) of marine animals in Thailand [54]. It is not clear what caused the arsenic contamination, which can come from several sources such as fertilizers, pesticides, and industrial effluent [55]. These biological contaminants may enter the human body through the food chain [23]. Therefore, there is a growing concern that it may cause harm to human health in the future.

### 3.5. Microplastic Accumulation in Odonate Larvae

A total of 419 individuals, belonging to three taxa of odonate larvae, were analyzed for MP presence (Table 5). The total number of MP particles in this study was 219 particles from all the species. Three shapes of MPs were recorded, namely, fibers, fragments, and rods. Fibers (80.82%) dominated the MP composition in all sampling stations, followed by fragments (18.26%) and rods (0.91%), respectively. The predominance of fibers in freshwater aquatic insects has been observed in prior studies [56,57,58], which were similar to our study. The MP particles ranged from less than 100 µm to more than 500 µm in size. The size category of less than 100 µm was the most common in odonate larvae. In terms of color, the MP particles observed under a stereomicroscope showed various colors, with the following predominance: transparent > violet > pink > blue > yellow and red (Table 6). Images of some recorded MPs and their corresponding polymer spectra are shown in Figure 3. In the odonate larvae, FTIR analysis revealed the presence of the following MP polymers: polyethylene terephthalate, polyvinyl acetate, bis(2-ethylhexyl), polybutadiene, poly(methyl methacrylate-co-methacrylic acid); P(MMA-co-MA), poly(ethylene glycol) tetrahydrofurfuryl ether, poly(acrylonitrile-co-butadiene), and polypropylene glycol. In the present study, the highest level of MP contamination was observed in Libellulidae (RF4) and Aeshnidae (RF5). Aeshnidae was observed to have a high level of contamination (12.00 MP items/individual). This was reasonable, as they are larger odonate species that showed a much higher MP load per individual, and body size can influence the rate of microplastic uptake [56]. However, a previous study reported that there was no correlation between the size of the freshwater macrobenthic invertebrates and the size or number of ingested MP particles [58]. These findings suggest that there is MP contamination in the aquatic insects that humans eat.

## 4. Conclusions

The nutritional composition of the aquatic insects that are consumed by humans has to date not been comprehensively determined. In this study, the nutrient composition and mineral contents of *Pantala* sp. showed that it is a source of particularly nutritional protein, fat, minerals, and amino acids, especially essential amino acids including valine, leucine, lysine, histidine, methionine, isoleucine, phenylalanine, and threonine. The fatty acid profile showed that it is a good source of n-6 PUFA and n-3 PUFA, e.g., linoleic acid (LA), arachidonic acid (AA), alpha-Linolenic acid, and eicosapentaenoic acid (EPA). Although insects have high nutritional value, they may typically contain As, Pb, and Cd contamination, including MPs, in their diets and habitats. Hence, more studies to evaluate their hazards need to be carried out to understand whether they may harm human health.

## Figures and Tables

**Figure 1 insects-13-01131-f001:**
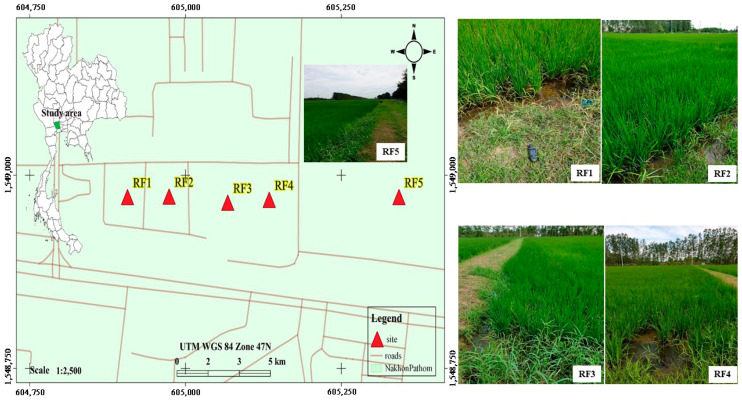
A map of five sampling sites was drawn using the QGis 3.14.1 program (https://www.qgis.org/en/site/, accessed on 9 August 2022).

**Figure 2 insects-13-01131-f002:**
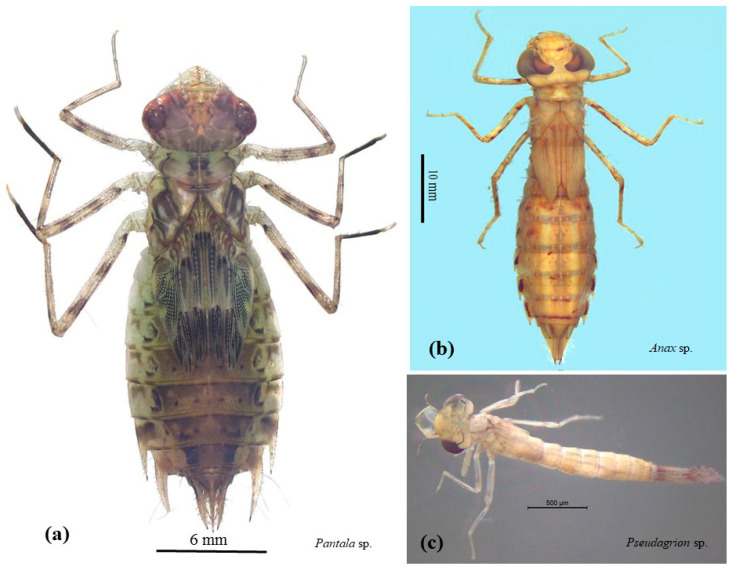
The characteristics of the odonatan larvae, (**a**) *Pantala* sp., (**b**) *Anax* sp., and (**c**) *Pseudagrion* sp., found in the rice fields.

**Figure 3 insects-13-01131-f003:**
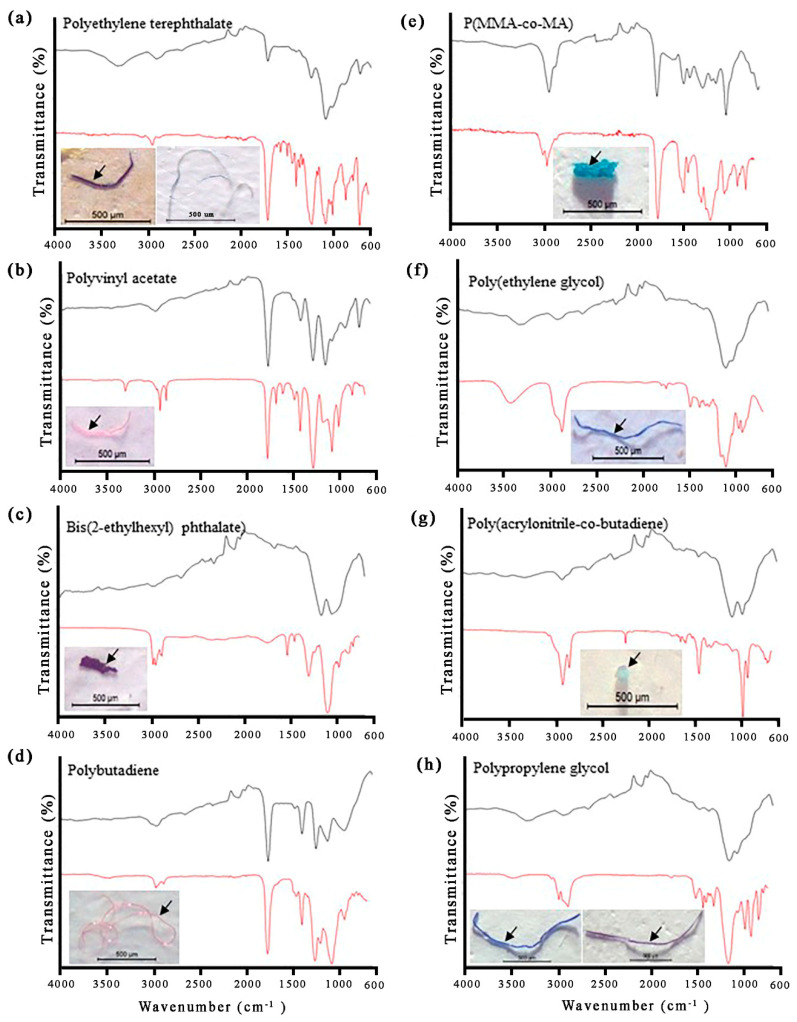
Representative MPs in odonate larvae and the FTIR spectra of MP polymers. The black spectrum is that of the FTIR measurement, while the red spectrum is the reference spectrum from the Bruker spectrum library.

**Table 1 insects-13-01131-t001:** Proximate compositions of odonate larvae, *Pantala* sp., g/100 g dry weight.

Parameter	Value (Means ± SD, *n* = 3)
Moisture	30.86 ± 0.04
Ash	5.61 ± 0.08
Fat	5.29 ± 0.09
Protein	49.45 ± 0.32
Total carbohydrate	8.80 ± 0.29
Total energy, (Kcal/100 g)	280.55 ± 0.28
Energy from fat, (Kcal/100 g)	47.58 ± 0.82

**Table 2 insects-13-01131-t002:** Fatty acid composition of odonate *Pantala* sp.

Saturated Fatty Acid (g/100 g)	
Lauric acid (C12:0)	0.02 ± 0.00
Myristic acid (C14:0)	0.10 ± 0.01
Pentadecanoic acid (C15:0)	0.14 ± 0.00
Palmitic acid (C16:0)	1.19 ± 0.02
Heptadecanoic acid (C17:0)	0.24 ± 0.01
Stearic acid (C18:0)	0.79 ± 0.02
Arachidic acid (C20:0)	0.10 ± 0.00
Behenic acid (C22:0)	0.07 ± 0.01
Total SFA	2.65
Unsaturated fatty acid (g/100 g)	
Palmitoleic acid (C16:1)	0.27 ± 0.01
cis-10-Heptadecenoic acid (C17:1n10)	0.14 ± 0.01
cis-9-Oleic acid (C18:1n9c)	0.63 ± 0.02
Total MUFA	1.04
Linoleic acid (C18:2n6c)	0.55 ± 0.01
gamma-Linolenic acid (C18:3n6)	0.04 ± 0.00
alpha-Linolenic acid (C18:3n3)	0.33 ± 0.01
cis-11,14-Eicosadienoic acid (C20:2)	0.02 ± 0.00
cis-8,11,14-Eicosatrienoic acid (C20:3n6)	0.02 ± 0.01
Arachidonic acid (C20:4n6)	0.30 ± 0.01
Eicosapentaenoic acid (C20:5n3)	0.16 ± 0.01
Total PUFA	1.42

Values are means ± SD (*n* = 3), MUFA (monounsaturated fatty acids); PUFA (polyunsaturated fatty acids).

**Table 3 insects-13-01131-t003:** Amino acid (AA) composition of odonate *Pantala* sp.

Non-Essential AA (g/100 g)	
Aspartic acid	2.92 ± 0.03
Glutamic acid	4.92 ± 0.07
Serine	1.68 ± 0.01
Glycine	1.89 ± 0.02
Alanine	3.21 ± 0.03
Proline	5.11 ± 0.50
Arginine	5.46 ± 0.05
Tyrosine	2.30 ± 0.01
Cystine	1.25 ± 0.01
**Essential AA (g/100 g)**	
Threonine	1.76 ± 0.01
Histidine	1.23 ± 0.01
Valine	2.73 ± 0.03
Methionine	0.85 ± 0.02
Isoleucine	1.63 ± 0.02
Phenylalanine	1.33 ± 0.02
Tryptophan	0.21 ± 0.00
Leucine	2.49 ± 0.03
Lysine	2.24 ± 0.03
Total amino acids	43.21
Total essential amino acids	14.47

Values are means ± SD (*n* = 3).

**Table 4 insects-13-01131-t004:** Mineral content (mg/100 g).

Mineral Content	Concentration
Sodium	654.30 ± 5.82
Calcium	284.64 ± 11.92
Iron	86.74 ± 4.29
Potassium	803.74 ± 6.12
Magnesium	87.59 ± 1.29
Phosphorus	615.76 ± 13.90
Zinc	6.18 ± 0.04
Copper	1.48 ± 0.01
Lead (mg/kg)	0.18 ± 0.01
Cadmium (mg/kg)	0.17 ± 0.00
Arsenic (mg/kg)	3.51 ± 0.12

Values are means ± SD.

**Table 5 insects-13-01131-t005:** Shape and size distribution of microplastics in odonate larvae from rice fields.

Sites	Odonate Larvae	No. *	Total MP Items	Shape			Size (µm)				MP Items/Individual
				Fiber	Fragment	Rod	<100	200–250	250–500	>500	
RF1	Coenagrionidae	95	15	14	1		3	3	1	8	0.16
	Libellulidae	49	30	24	6		21	3	4	2	0.61
RF2	Coenagrionidae	113	23	23			15	4	3	1	0.20
	Libellulidae	48	44	36	8		21	11	6	6	0.92
RF3	Coenagrionidae	53	12	11	1		2	4	2	4	0.23
	Libellulidae	44	22	7	13	2	15	3	2	2	0.50
RF4	Coenagrionidae	7	33	24	9		19	6	4	4	4.71
	Libellulidae	1	17	16	1		6	5	4	2	17.00
RF5	Libellulidae	8	11	10	1		5	4	2		1.38
	Aeshnidae	1	12	12			4	3	3	2	12.00
Total (%)			219 (100)	177 (80.82)	40 (18.26)	2 (0.91)	111 (50.68)	46 (21.00)	31 (14.16)	31 (14.16)	

* No. of larvae pooled for analysis.

**Table 6 insects-13-01131-t006:** Color distribution of microplastics in odonate larvae from rice fields.

Sites	Odonate Larvae	Color						Total MPs
		Blue	Violet	Pink	Transparent	Yellow	Red	
RF1	Coenagrionidae	3		3	8		1	15
	Libellulidae	1	2	6	21			30
RF2	Coenagrionidae		3		20			23
	Libellulidae	1	13	3	27			44
RF3	Coenagrionidae	1	6	1	4			12
	Libellulidae	2	14	1	5			22
RF4	Coenagrionidae		8	3	22			33
	Libellulidae	3	3	1	10			17
RF5	Libellulidae	1			9	1		11
	Aeshnidae	2	1		9			12
Total (%)		14 (6.39)	50 (22.83)	18 (8.22)	135 (61.64)	1 (0.46)	1 (0.46)	219

## Data Availability

The data presented in this study are available on request from the corresponding author.

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
