# Peer review of "Nutritional Compositions of Aquatic Insects Living in Rice Fields, with a Particular Focus on Odonate Larvae"

_insects, 2022, doi:10.3390/insects13121131_

Round 1

Reviewer 1 Report

I liked the manuscript overall. However, I have several issues to be solved before it can be published. First, methods are not described in a comprehensive manner. For instance, we have no information about the number of specimens analyzed (in nutritional analyses, I guess it might have been three specimens – based on Table 1 – if so, I am afraid it is not enough. If it wasn’t that way, then your methods are not well described). Second, I miss a proper discussion at some points (see the detailed comments below). Finally, professional English check is more than needed.

18 – you use the abbreviation “MP” here for the first time but without any explanation

132 – how many specimens? Have you any idea of the instar used for the analysis? Based on the pictures I guess it is a final or pre-final instar, but did you have any other, “younger” specimens as well?

135 – you mean minus 25 °C, right? (If so, then maybe you meant they were kept in a freezer)

137–138 “The proximate analysis was estimated following the method of the AOAC techniques 925.45, 991.20, 2003.05, 138 and 923.03” What are these methods? Could you provide a brief description or reference them for the unexperienced readers?

150 – How many specimens? I suppose different individuals (different form those used for nutritional analysis) were used for this analysis of microplastics – you used Pantala sp. (well, I see it now in the results but it should really be mentioned in the methods) for nutritional analyses, and various species for analysis of MP, right? If so, please make it clear. What species did you have in the analysis of MPs? Did you determine only the family?

182–183 To what extent are Pantala sp. larvae consumed? Why did you choose to analyze this species? I miss some discussion or introduction on this topic; how frequent eating odonate larvae is (in Vietnam or in other parts of the world)? I suppose rice fields are more heavily impacted by heavy metals and possible bioaccumulation effect than other type of habitats. Is consummation of odonate larvae from rice fields a common practice?

245 – the reported values of iron in Pantala are six times higher than in Sympetrum which is weird. Do you have any idea why? For example, could it result from the differences in the habitat both species occupy (e.g., were Sympetrum larvae also collected at rice fields)? It would be nice to have it discussed in the paper.

263–265 “Similarly, Aydogan et al. [16] also reported levels of arsenic in some aquatic insects (Coleoptera) had high arsenic levels (0.2-14.4 mg·kg-1) that were higher than our results” This sentence seems weird; please, reformulate.

261–265 Again, do you have any idea why were aquatic Coleoptera higher in arsenic content? I imagine it could be affected by the fact that you analyzed larvae whereas those aquatic Coleoptera were probably adults (longer-lived in the water = more arsenic accumulated from food, etc.). Or, maybe, there are some physiological differences among Odonata and Coleoptera critically affecting their bioaccumulation. Again, the discussion on this topic would be beneficial.

267–268 What about some reference levels of heavy metals in human food – what are the levels of heavy metals critical for human health? Are the levels you found in Pantala sp. really a threat?

291–292 “In the present study, the highest level of contamination was observed in predators feeding on other consumers, which feed on small organisms or material sedimented.” I don’t understand this sentence. In your study, you analyzed only odonate larvae, so only predators feeding on other consumers.

Author Response

-I tried to answer all your questions by taking advice from the revised manuscript. In this regard, I have attached the revised file from the experts and this manuscript, which I have sent for English editing and proofreading (certificates are provided by MDPI).

Reviewer 2 Report

Please check the comments on the manuscript

Author Response

(The authors gave the same response as above.)

Round 2

Reviewer 1 Report

The authors have answered all my comments / concerns.